# The risk of bias in denoising methods: Examples from neuroimaging

**Kendrick Kay**⊙*

Department of Radiology, Center for Magnetic Resonance Research (CMRR), University of Minnesota, Minneapolis, MN, United States of America

* kay@umn.edu

## Abstract

Experimental datasets are growing rapidly in size, scope, and detail, but the value of these datasets is limited by unwanted measurement noise. It is therefore tempting to apply analysis techniques that attempt to reduce noise and enhance signals of interest. In this paper, we draw attention to the possibility that denoising methods may introduce bias and lead to incorrect scientific inferences. To present our case, we first review the basic statistical concepts of bias and variance. Denoising techniques typically reduce variance observed across repeated measurements, but this can come at the expense of introducing bias to the average expected outcome. We then conduct three simple simulations that provide concrete examples of how bias may manifest in everyday situations. These simulations reveal several findings that may be surprising and counterintuitive: (i) different methods can be equally effective at reducing variance but some incur bias while others do not, (ii) identifying methods that better recover ground truth does not guarantee the absence of bias, (iii) bias can arise even if one has specific knowledge of properties of the signal of interest. We suggest that researchers should consider and possibly quantify bias before deploying denoising methods on important research data.

## Introduction

Modern science has witnessed major advances in the application of computational analyses to large datasets [1, 2]. This has led to a 'big data' revolution in which datasets of increasing size, scope, and detail are being amassed [3–5]. In the field of neuroscience, advances in electrophysiological, optical, and magnetic resonance techniques are enabling measurement of the structure and function of animal and human brains at higher resolution, with greater coverage, and over longer temporal durations. However, a major challenge in these measurements is the presence of noise, which we define as unwanted variability across repeated measurements from the same individual. Such noise can originate from a variety of sources and can be both structured (e.g., imaging artifacts, head motion, physiological noise, variations in cognitive performance) and unstructured (e.g., thermal noise, optical shot noise). Depending on the goals of a given experiment, many of these types of noise are undesirable to the researcher.

Developing methods for removing noise from data has been a long-standing objective in neuroscience. High levels of noise in experimental data hinder scientific inferences; thus, there

**Data Availability Statement:** The data and code used to carry out the simulations in this paper are available on the Open Science Framework (https://osf.io/weg87/). These materials can be easily adapted to explore different simulation scenarios (different noise levels, different ground truths, etc.).

**Funding:** The author received no specific funding for this work.

**Competing interests:** The authors have declared that no competing interests exist.

is a temptation to apply denoising methods to such data. Indeed, there are many interesting recently proposed approaches for denoising, including low-rank methods [6–8], methods based on data-driven noise derivation [9–11], methods that exploit the power of deep neural networks [12–15], and self-supervised methods [16]. In surveying the literature, we find extensive discussion and consideration of denoising methods and how they fare in specific scientific paradigms. However, we think that, aside from a few notable exceptions [17, 18], there has been insufficient emphasis on the issue of statistical bias.

Bias, in the statistical sense, is defined as the discrepancy between the average expected outcome of a given experiment (and its associated analysis) and the ground-truth parameter being estimated (a more formal treatment is provided later). In expositions of denoising methods, the possibility of bias is often not even mentioned or discussed, let alone quantified and assessed. Coming to clarity on this methodological issue is especially important in the context of modern datasets. This is because increasing sizes of datasets, increasing levels of noise (due to increased spatial resolution, temporal resolution, and acquisition speeds), and increasing complexity of data analysis pipelines all tend to obscure or make more difficult the assessment of bias. A critical message of this paper is that bias is risky: while a method might improve the correspondence between a noisy dataset and a ground-truth measure, this might come at the cost of introducing systematic biases into the data and lead to incorrect scientific inferences.

We write this article with two goals in mind. First, we wish to draw attention to—or perhaps rekindle interest in—the basic statistical concepts of bias and variance. Our presentation is general in order to isolate the essential principles at stake. We attempt to provide a concise distillation of the concepts of bias and variance that is easy to understand for non-statisticians. Second, we wish to communicate several simulations that illustrate how these concepts and principles can be applied in concrete scientific paradigms. We design these examples based on our experience in neuroimaging, and we make freely available the underlying data and code to promote transparency (files available at https://osf.io/weg87/). The examples are not intended to establish general methodological findings (for that, more extensive analyses are necessary), but rather to provide important insights into the nature of denoising. We acknowledge that the ideas and principles we convey may already be apparent to expert practitioners. Thus, perhaps the primary audience of this paper are researchers who are interested in—but have not fully developed their stance towards—strategies for denoising data. Ultimately, we hope this article spurs method developers to consider and potentially quantify bias in candidate denoising methods and users to consider the risk of bias when applying denoising methods to important research data.

## Materials and methods

### Simulation framework

All simulations (as depicted in Figs 2–4) use a common analytical framework. We first design a ground truth based on either empirical or synthetic data. We then generate simulated data by adding randomly generated noise to the ground truth. This produces a set of measurements, each of which may contain multiple data points (e.g. different voxels, different time points). Next, we apply various denoising methods. Each method is applied independently to each measurement and produces a set of analysis results. Finally, for each method, we compute quantitative metrics that assess the performance of the method. Three metrics are computed and are detailed below.

*Bias* is quantified by computing, for each data point, the absolute deviation between the mean across analysis results and the ground truth, normalized by the standard error across analysis results (this normalization can be viewed as a form of studentization, in which a

quantity is normalized by a measure of error, producing units that are easy to interpret). Note that computing the absolute value is important, since a denoising method might overestimate and underestimate the ground truth in different parts of a dataset and it should be penalized for doing so. We summarize the results by calculating the median absolute deviation across data points. The values are in normalized units, and low values are desirable, as they indicate low deviations from ground truth. Data points for which the standard error across analysis results is 0 are ill-defined and are ignored in the calculation (e.g. Fig 3B, right column, time = 0 s).

It is important to note that our metric of bias is not, strictly speaking, the same as the idealized theoretical definition of statistical bias (see Eq 1). The theoretical definition would require computing expectation over an infinite (or very large) number of simulations; in contrast, our metric is suitable for computation in finite data regimes and takes into account the limited number of simulations through normalization by standard error (with the underlying idea that running more simulations to reduce standard error is always desirable, if computational resources are available). One issue with the metric is that non-zero values are obtained even for unbiased measurements (thus, the metric can be viewed as the "apparent bias"). Therefore, to provide a suitable comparison, we perform Monte Carlo simulations (assuming a Gaussian noise distribution) to determine the value that is expected for the case of unbiased measurements; this value is plotted as 'Baseline' in Figs 2–4. Note that this baseline value can be computed analytically as tinv(0.75,$v$) which indicates the inverse of the cumulative distribution function associated with Student's $t$-distribution, evaluated at 75% and $v$ degrees of freedom. For example, in the case of 10 measurements, tinv(0.75,9) = 0.70 indicating that half of a set of samples drawn from a $t$-distribution with 9 degrees of freedom are expected to have an absolute value less than or equal to 0.70.

*Variance* is quantified by computing, for each data point, the standard error across analysis results. We summarize the results by calculating the median standard error across data points. The values are in the units of the original data, and low values are desirable, as they indicate high reliability of analysis results.

*Error* is quantified by computing Pearson's correlation between each analysis result and the ground truth. (Note that correlation allows flexibility for scaling and offset; while a non-flexible metric such as mean squared error is technically more correct, correlation is appealing for its interpretable units and is likely sufficient in most cases.) We summarize the results by calculating the mean correlation observed across analysis results. Intuitively, this metric assesses how well a denoising method recovers ground truth. Correlation values range from –1 to 1. High values are desirable, as they indicate high similarity of analysis results to the ground truth.

## Simulation 1: Anatomical data

In this simulation, we use as ground truth the pre-processed 0.8-mm $T_1$-weighted anatomical volume acquired from Subject 1 from the Natural Scenes Dataset (NSD) [19]. The intensity values in this volume range approximately from 0 to 1400 (see Fig 2A, middle). Also from NSD, we use the brain mask calculated for the subject and the tissue segmentation provided by FreeSurfer (see Fig 2A, bottom). We map the 1-mm MNI $T_1$-weighted atlas provided with FSL (https://fsl.fmrib.ox.ac.uk/fsl/) to the subject-native anatomical space using linear interpolation (see Fig 2A, top). We generate a set of 10 measurements by adding noise drawn from a Gaussian distribution with mean zero and standard deviation 300 (noise drawn independently for each voxel). We evaluate four denoising methods: (1) *No denoising* refers to using the measurements as-is. (2) *Gaussian smoothing* refers to spatially smoothing a given measurement

using a 3D isotropic Gaussian kernel with a full-width-half-maximum (FWHM) of 3 mm. (3) *MNI atlas prior* refers to averaging a given measurement with the MNI atlas (mapped to subject-native space). Before averaging, a scale and offset is applied to the atlas such that the mean of the data within gray matter (as indicated by the tissue segmentation) and the mean of the data within white matter are matched to the corresponding gray- and white-matter means in the MNI atlas. (4) *Anisotropic smoothing* refers to applying nonlinear anisotopic diffusion-based smoothing [20] as implemented in Segmentator [21]. The diffusion-based smoothing is run for 20 iterations. For all denoising methods, quantitative metrics of performance (as described previously) are computed using voxels within the brain mask.

## Simulation 2: Response timecourses

In this simulation, we use as ground truth a synthetic hemodynamic response function (HRF) generated by evaluating a double-gamma function as implemented in SPM's spm_hrf.m (https://www.fil.ion.ucl.ac.uk/spm/). The parameters [6 16 1 1 2 0] are used; these are the defaults, except for the fifth parameter, which is set to create a strong undershoot. The double-gamma function is convolved with a 1-s boxcar, sampled at a rate of 1 s, and then scaled to peak at 1. The resulting HRF represents a hypothetical fMRI response timecourse to a 1-s stimulus (see Fig 3A, top). We generate a set of 10 measurements by adding temporally correlated Gaussian noise with mean zero and standard deviation 0.2 (this was accomplished by generating zero-mean Gaussian noise with standard deviation 0.2 and convolving the noise with a 5-s boxcar scaled to have a Euclidean norm of 1). We evaluate three denoising methods: (1) *No denoising* refers to using the measurements as-is. (2) *Basis restriction* refers to projecting the measurements onto a set of basis functions and then reconstructing the measurements. For basis functions, we take the library of 20 canonical HRFs obtained from the Natural Scenes Dataset [19] (getcanonicalhrflibrary.m), predict the response to a 1-s stimulus, perform principal components analysis on the 20 timecourses, and extract the top three principal component timecourses (see Fig 3A, bottom). (3) *Parametric fit* refers to fitting each measurement using a double-gamma model (same as used to generate the data). Specifically, we use nonlinear optimization (MATLAB Optimization Toolbox's lsqnonlin.m) to determine the optimal parameters for a double-gamma function (as implemented in SPM's spm_hrf.m) such that when convolved with a 1-s boxcar, the result best approximates the measurement in a least-squares sense. The initial seed for the optimization is set to [6 16 1 1 6 0], which are the defaults in spm_hrf.m.

## Simulation 3: Tuning curves

In this simulation, we use as ground truth a synthetic set of tuning curves associated with several hypothetical units (these units can be thought of as individual neurons or voxels). We construct tuning curves that represent the response of 10 units to 50 conditions—these conditions can be viewed as different points along some hypothetical stimulus dimension. We fix the dimensionality of the representation to be exactly 4. This is accomplished by creating 4 Gaussian functions spaced equally along the stimulus dimension, and then generating tuning curves for each unit by weighting and summing these Gaussian basis functions using a set of randomly generated weights (random numbers are drawn from a uniform distribution between 0 and 1 and then cubed). Each unit's tuning curve is scaled to peak at 1, and to aid visibility, units are arranged in sorted order according to the center-of-mass of each tuning curve (see Fig 4A). We generate a set of 30 measurements by adding noise drawn from a Gaussian distribution with mean zero and standard deviation 0.6. (For visibility, only 10 of these 30 measurements are shown in Fig 4B, bottom row.) We evaluate three denoising methods. (1) *No*

*denoising* refers to using the measurements as-is. (2) *Boxcar smoothing* refers to smoothing each unit's measured tuning curve using a boxcar kernel with width 3 and integral 1 (this is simply a moving average with window size 3). (3) *PCA* refers to reducing the dimensionality of each measurement to a specific target rank, a method also referred to as Truncated SVD [22]. Variants of this method can be found in the literature [6, 7]. Specifically, given a measurement $X$ (10 units × 50 conditions), we perform singular value decomposition to obtain $X = USV^T$ where $U$ (10 × 10) has loadings in the columns, $S$ (10 × 50) has singular values in decreasing order on the diagonal and zeros elsewhere, and $V$ (50 × 50) has timecourse components in the columns. We then perform low-rank reconstruction of the measurement using $n$ = 2, 3, 4, 6, or 8 components (referred to as PCA2, PCA3, PCA4, PCA6, and PCA8) by computing the reconstructed measurement $X^* = U^* S^* V^{*T}$ where $U^*$ contains the first $n$ columns of $U$, $S^*$ contains the upper-left $n \times n$ elements of $S$, and $V^*$ contains the first $n$ columns of $V$.

### Tissue segmentation

To provide an example of the downstream impact of denoising, we carry out post-hoc analyses on the results of the first simulation (anatomical data). First, we generate a noisy measurement using a noise level of standard deviation 100. We then apply the four denoising methods (as previously described) to the measurement. Given that a typical goal in anatomical imaging is to identify different anatomical structures, we attempted to segment the data produced by each denoising method. Specifically, we take each result, perform skull stripping using FSL's BET (Brain Extraction Tool), and then use FSL's FAST (FMRIB's Automated Segmentation Tool) [23] to obtain a tissue segmentation. In Fig 5, we show the hard segmentation output ('seg') which provides labels for cerebrospinal fluid, gray matter, and white matter.

## Results

### A brief review of bias and variance

We start by briefly reviewing some basic statistical concepts [24, 25]. Suppose we are interested in estimating a certain population parameter by performing measurements of this parameter. There are two distinct aspects of the quality of our measurements: bias and variance. *Bias* refers to the discrepancy, if any, between the average expected outcome of our measurements and the population parameter. All else being equal, we want bias to be zero (or low), since we want our measurements to cluster around the true value of the population parameter. *Variance* refers to the variability of our measurements. All else being equal, we want variance to be low, since this helps us narrow down a range of plausible values for the population parameter.

A simple example helps illustrate these concepts. Fig 1 depicts a 2 × 2 crossing of different measurement scenarios. The columns differ in the amount of measurement bias. The left column corresponds to unbiased measurement, in which measurements, on average, equal the ground-truth parameter, whereas the right column corresponds to biased measurement, in which measurements, on average, are higher than the ground-truth parameter. The rows differ in the amount of measurement variance. The top row corresponds to low-variance measurement, in which measurements cluster tightly together, whereas the bottom row corresponds to high-variance measurement, in which measurements are spread far apart.

A common approach for assessing how well a measurement procedure captures the population parameter is to compute *mean squared error* (MSE), which refers to the average squared deviation of the measurements from the population parameter. It is important to note that this error metric reflects *separate contributions of bias and variance*. Specifically, mean squared

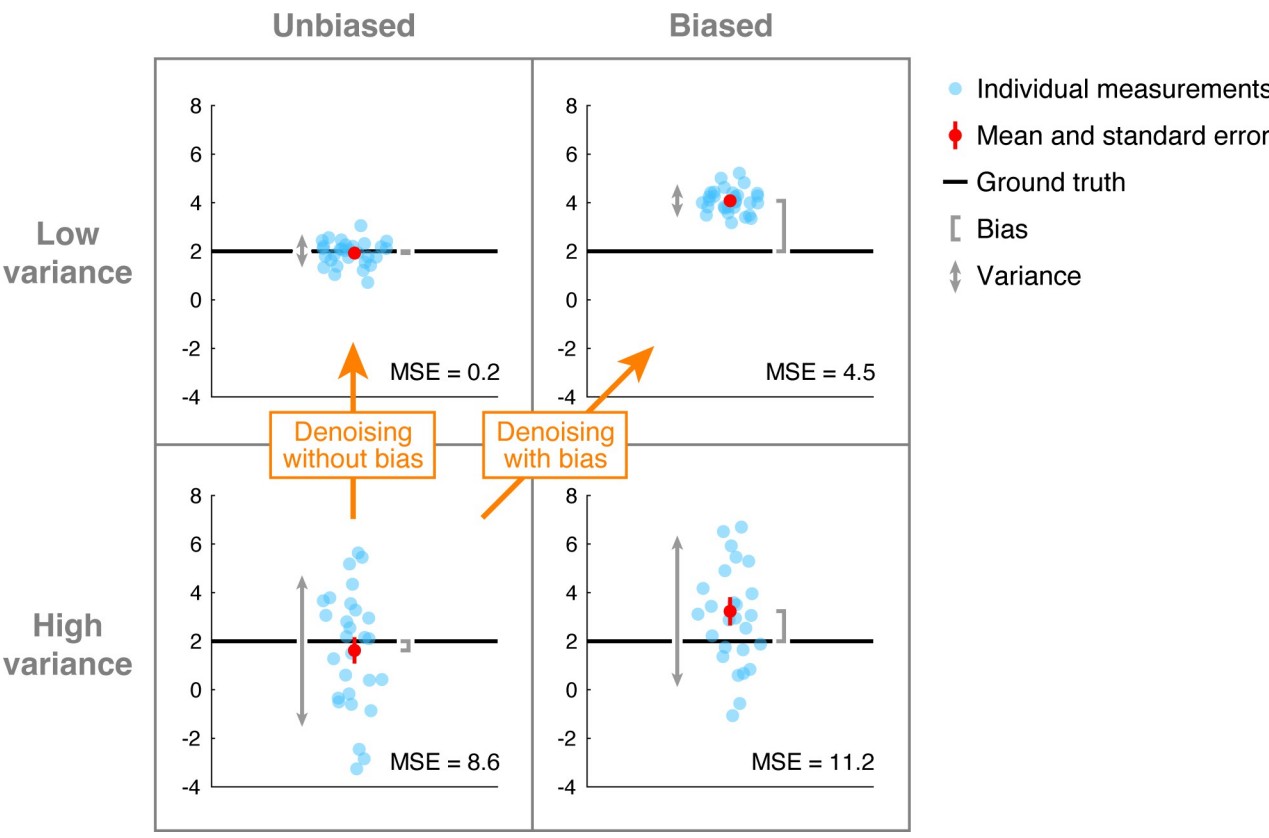

**Fig 1. Bias and variance in measurement.** In each of the four depicted simulations, 2 is the ground-truth value and 30 measurements are simulated by drawing values from a Gaussian distribution. In the left column, the Gaussian distributions have a mean of 2 (unbiased), whereas in the right column, the distributions have a mean of 4 (biased). In the top row, the Gaussian distributions have a variance of 0.3 (low variance), whereas in the bottom row, the distributions have a variance of 8 (high variance). The inset indicates the mean squared error (MSE) between the measurements and the ground truth. Bias can be estimated as the discrepancy between the mean of the measurements and the ground truth. Variance can be estimated as the variability across the measurements. Code available at https://osf.io/6x8kq/.

error is equal to the sum of two separate terms, a squared-bias term and a variance term:

$$\mathrm{MSE} = \mathrm{BIAS}^2 + \mathrm{VARIANCE} \tag{1}$$

To see why this is the case, we first define bias as the difference between the average expected measurement and the ground-truth value:

$$\mathrm{BIAS} = \mathbb{E}[\hat{y}] - y \tag{2}$$

where $y$ indicates the ground-truth value, $\hat{y}$ indicates a single measurement, and $\mathbb{E}$ is the expectation operator indicating the average over an infinite number of repeated measurements. We compute the squared bias as follows:

$$\mathrm{BIAS}^2 = y^2 - 2y\mathbb{E}[\hat{y}] + \left(\mathbb{E}[\hat{y}]\right)^2 \tag{3}$$

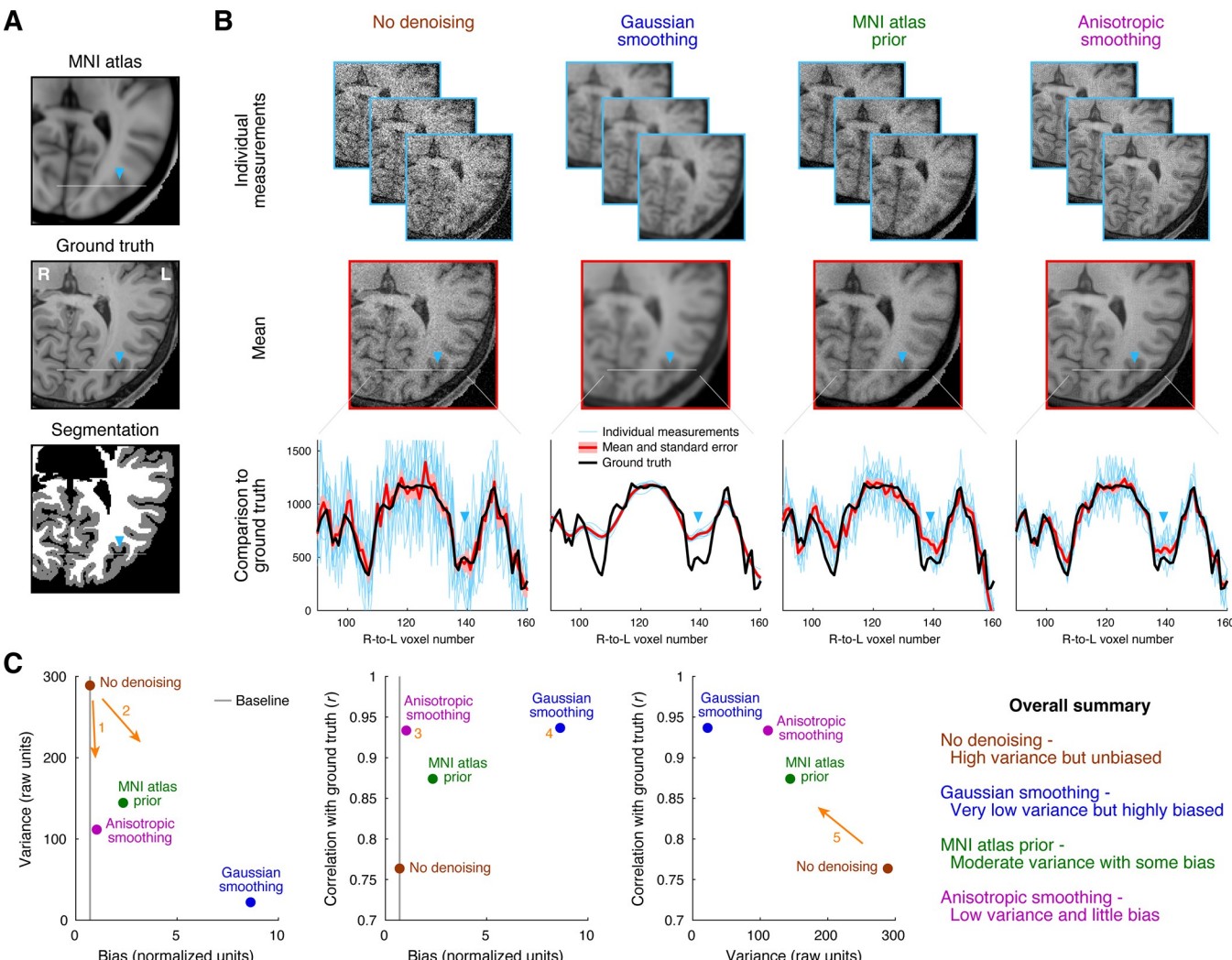

**Fig 2. Denoising anatomical data.** In this simulation (code available at https://osf.io/qxp8y/), we generate noisy measurements by starting with a ground-truth $T_1$-weighted anatomical volume and adding Gaussian noise independently to each voxel. We then attempt to denoise the data using different denoising methods: no denoising, simple Gaussian spatial smoothing, averaging with a group-average atlas prior, and performing anisotropic diffusion. The images depict a zoomed-in view of the posterior section of a single axial slice, and the same color map and range is used for all images. (A) Reference volumes. We illustrate the ground-truth anatomical volume (middle), the MNI atlas used in one of the denoising methods (top), and the tissue segmentation obtained from FreeSurfer, showing gray and white matter (bottom). (B) Denoising results. Each column shows results for a different denoising method. We show three example measurements (top row), the mean across measurements (middle row), and detailed plots for a small line of voxels (bottom row). (C) Quantitative assessment of bias, variance, and error. Bias is quantified as the median absolute difference between the average measurement and the ground truth, where the difference is normalized by the standard error across measurements. Variance is quantified as the median standard deviation across measurements. Error is quantified as the correlation between each measurement and the ground truth, averaged across measurements. The gray vertical line indicates the bias value associated with the case of unbiased measurement (assuming Gaussian noise).

Note that squared bias is always non-negative. Next, we define variance as the average squared deviation of the measurements around their mean:

$$\begin{aligned}
\text{VARIANCE} &= \mathbb{E}[(\hat{y} - \mathbb{E}[\hat{y}])^2] \\
&= \mathbb{E}[\hat{y}^2 - 2\hat{y}\mathbb{E}[\hat{y}] + (\mathbb{E}[\hat{y}])^2] \\
&= \mathbb{E}[\hat{y}^2] - 2\mathbb{E}[\hat{y}]\mathbb{E}[\hat{y}] + (\mathbb{E}[\hat{y}])^2 \\
&= \mathbb{E}[\hat{y}^2] - (\mathbb{E}[\hat{y}])^2
\end{aligned}$$
(4)

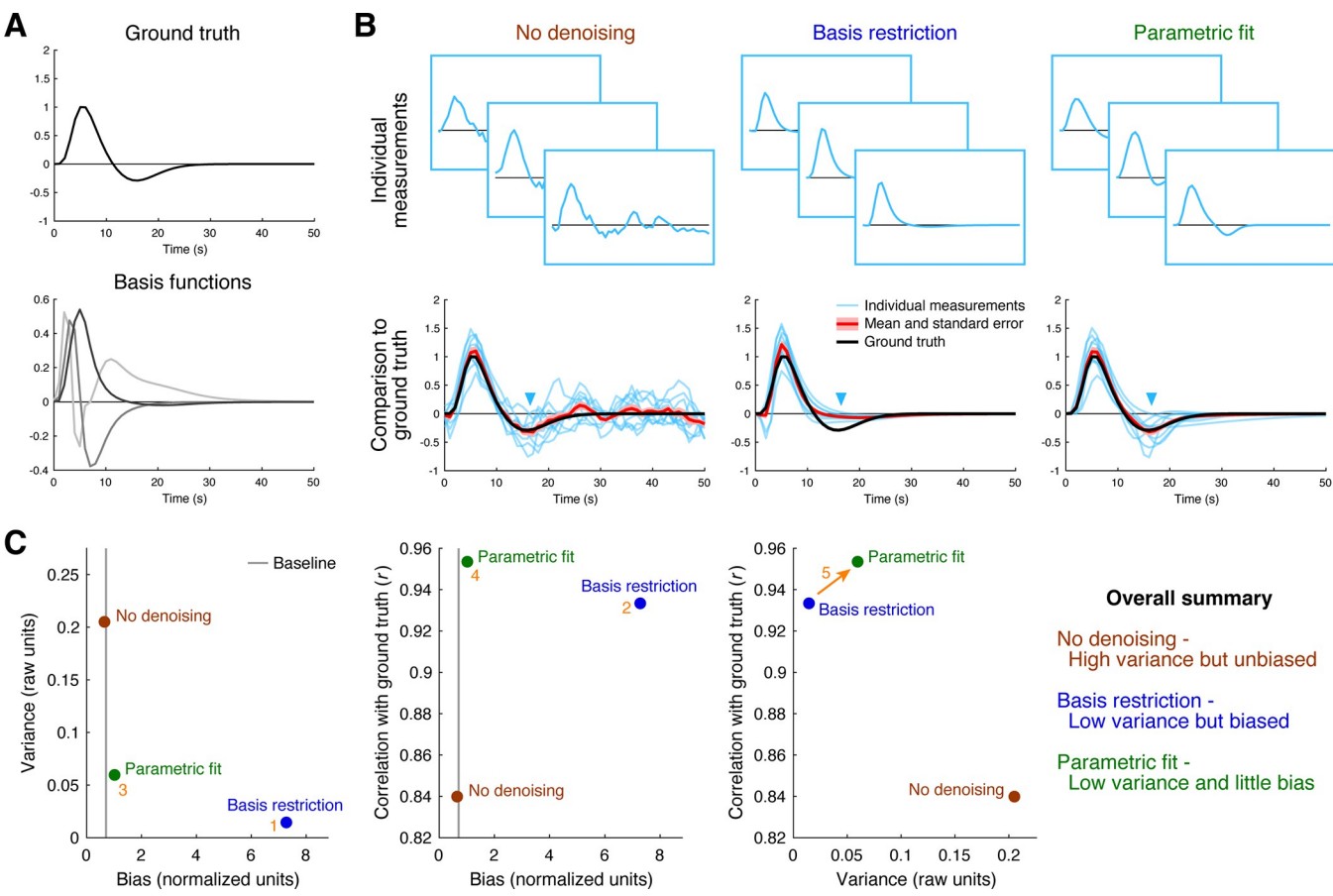

**Fig 3. Denoising response timecourses.** In this simulation (code available at https://osf.io/6jhmr/), we generate noisy measurements by starting with a ground-truth hemodynamic response function (HRF) and adding temporally correlated Gaussian noise. We then attempt to denoise the data using different denoising methods: no denoising, reconstruction using a restricted set of basis functions, fitting using a parametric model. (A) Reference timecourses. We illustrate the ground-truth HRF (top) and the temporal basis functions used in one of the denoising methods (bottom). (B) Denoising results. Each column shows results for a different denoising method. We show three example measurements (top row) and comparison to the ground truth (bottom row). (C) Quantitative assessment of bias, variance, and error. Same format as Fig 2C.

Finally, we define mean squared error as the average squared deviation of the measurements from the ground-truth value:

$$\begin{aligned} \text{MSE} &= \mathbb{E}[(y - \hat{y})^2] \\ &= \mathbb{E}[y^2 - 2y\hat{y} + \hat{y}^2] \\ &= y^2 - 2y\mathbb{E}[\hat{y}] + \mathbb{E}[\hat{y}^2] \end{aligned} \tag{5}$$

Adding some terms and grouping, we obtain:

$$\text{MSE} = (y^2 - 2y\mathbb{E}[\hat{y}] + (\mathbb{E}[\hat{y}])^2) + (\mathbb{E}[\hat{y}^2] - (\mathbb{E}[\hat{y}])^2) \tag{6}$$

By substituting from Eqs 3 and 4, we see:

$$\text{MSE} = \text{BIAS}^2 + \text{VARIANCE} \tag{7}$$

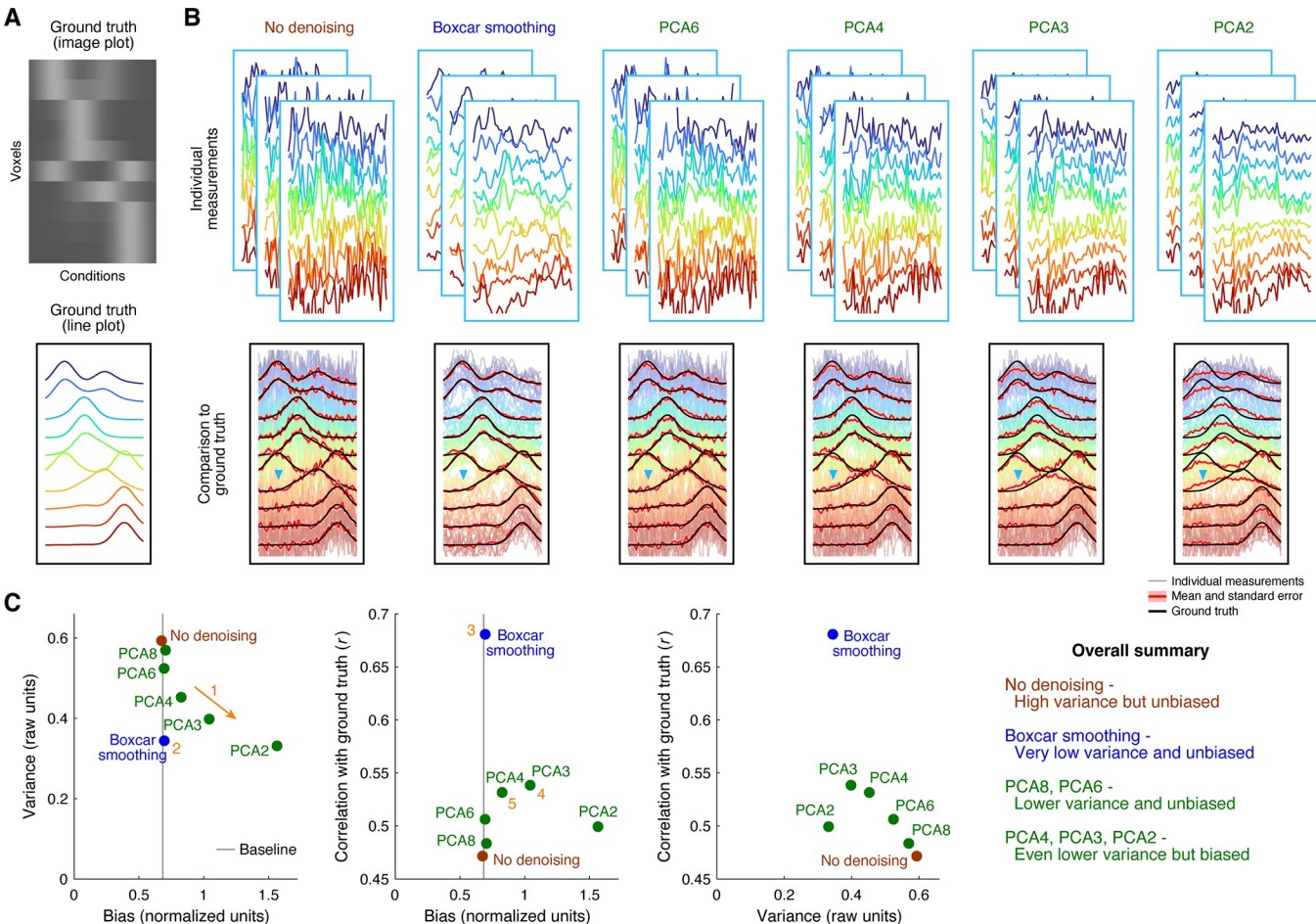

**Fig 4. Denoising tuning curves.** In this simulation (code available at https://osf.io/a6k9m/), we generate noisy measurements by starting with a ground-truth collection of tuning curves whose underlying dimensionality is fixed at 4 and adding Gaussian noise independently to each data point. We then attempt to denoise the data using different denoising methods: no denoising, simple boxcar smoothing of responses to nearby conditions, and dimensionality reduction using principal component analysis (PCA). (A) Reference data. We illustrate the ground-truth tuning curves as an image (top) and as line plots (bottom). Color is used to distinguish different units. (B) Denoising results. Each column shows results for a different denoising method. We show three example measurements (top row) and comparison to the ground truth (bottom row). (C) Quantitative assessment of bias, variance, and error. Same format as Fig 2C.

## Insights and implications for denoising

Having reviewed the concepts of bias and variance, we highlight some important insights. First, we remind ourselves of the classic distinction between reliability and accuracy. Even though a procedure might provide highly reliable measurements (low variance), this does not necessarily imply that that the measurements are accurate. This is because the measurements might have systematic deviation (bias) from the ground-truth parameter (e.g., see upper-right panel of Fig 1). Second, we observe that assessing error relative to ground truth does not provide specific information regarding bias. Error, as discussed earlier, reflects the combination of both bias and variance. Hence, a situation in which error is low is compatible with the existence of bias (e.g., in Fig 1, the upper-right panel exhibits lower error than the lower-left panel but has substantial bias).

We now transition to the topic of denoising. A common situation that an experimentalist may face is one in which a set of measurements are corrupted by high levels of noise but are at least expected to converge, across repeated experiments, to the true signal. This situation can

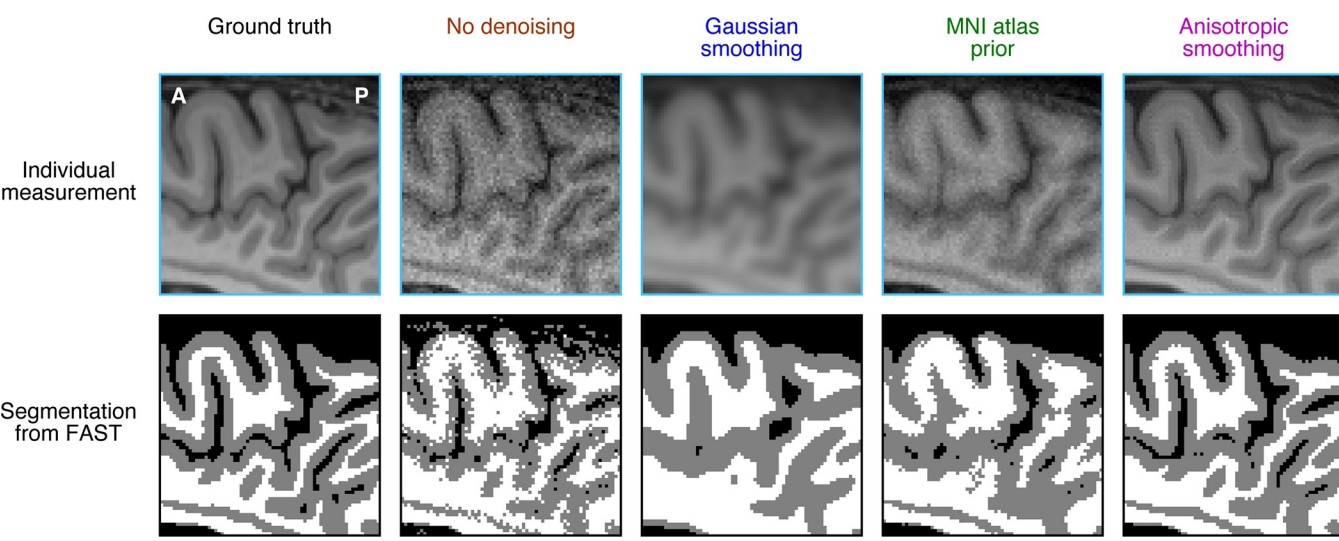

**Fig 5. Example downstream effects of denoising anatomical data.** Here, we perform post-hoc analyses on the results of the denoising methods illustrated in Fig 2 (code available at https://osf.io/hswaq/). Specifically, we simulate an example noisy measurement, apply different denoising methods, and calculate a tissue segmentation using FSL's FAST. The images depict a zoomed-in view of a superior section of a single sagittal slice. The first row shows the original data and the second row shows segmentation results (black, dark gray, and light gray indicate cerebrospinal fluid, gray matter, and white matter, respectively).

be characterized as high variance and unbiased (Fig 1, lower left). To reduce noise, the experimentalist might try applying a denoising technique to the data. In doing so, there are two general types of outcomes. One outcome is that variance is reduced while the absence of bias is maintained (see arrow labeled 'Denoising without bias' that begins in the lower-left panel and ends in the upper-left panel). This is a great outcome. A different outcome is that variance is reduced but bias is introduced (see arrow labeled 'Denoising with bias' that begins in the lower-left panel and ends in the upper-right panel). This is a less desirable outcome, as repeated experiments converge to an incorrect signal. Reduction of variance but introduction of bias is an instance of the classic bias-variance tradeoff [25]. From a certain perspective, one might argue that introducing bias is desirable if this reduces overall error [26]. However, we feel that this is risky and warrants careful consideration (see Discussion).

### Examples of bias and variance in denoising

While we have described theoretical considerations to take into account when assessing a denoising method, it may be unclear how much these considerations actually matter in practical situations. To provide more concrete insights, we construct three denoising simulations based on our experience with neuroimaging data. The goal of these simulations is to provide examples of how the performance of different denoising methods can be formally evaluated. In each example, we start with a ground truth, generate noisy measurements based on this ground truth, apply different denoising methods to each measurement, and calculate metrics that quantify the performance of the denoising methods. We generally follow the theory presented earlier, but use versions of the metrics that are more suitable and interpretable for practical data scenarios. Specifically, we quantify *bias* as the median absolute deviation between the mean across analysis results and the ground truth and express this in units of standard error; we quantify *variance* as the median standard error across analysis results; and we quantify *error* as the average correlation between each analysis result and the ground truth (see Methods). Please note that the denoising methods demonstrated in the examples are not intended to be realistic methods that one might want to use in practice (e.g., Gaussian smoothing is

obviously a naive approach; averaging with an MNI atlas is obviously a very crude approach). This is because the point of the examples is not so much to determine the best state-of-the-art denoising method, but rather to demonstrate how bias and variance can be formally studied.

In the first simulation, we use as ground truth a high-quality 0.8-mm isotropic anatomical MRI scan of a human brain (Fig 2A) and simulate noisy measurements of this ground truth by adding Gaussian noise. (Real noise in MRI data is better characterized according to Rician and/or other types of distributions, and may have complex spatial variations across the image [27, 28]. Here we use the simplifying assumption of Gaussian noise, and acknowledge that results may vary in interesting ways for other types of noise.) As expected, the raw data ('No denoising') follow the ground truth, in the sense of lacking bias, but suffer from high variance (Fig 2B, first column). The method of spatial smoothing ('Gaussian smoothing') reduces variance, but incurs major deviations from ground truth (Fig 2B, second column). This is not surprising since the smoothing kernel used has a relatively large full-width-half-maximum of 3 mm, which will obviously remove fine-scale features of the convoluted cerebral cortex. The method of averaging a given measurement with a pre-existing atlas ('MNI atlas prior') provides some variance reduction, but also introduces some bias (Fig 2B, third column). This makes sense, since the atlas is generally expected to provide good guesses for tissue intensity, but may bias the measurement in parts of the individual's brain that deviate from the atlas. Finally, the method of applying anisotropic smoothing ('Anisotropic smoothing') greatly reduces variance and, appealingly, introduces very little bias, if any (Fig 2B, fourth column). Our interpretation is that the assumption embodied by anistropic smoothing—namely, that true structures are locally contiguous and have homogeneous signal intensity—is well matched to the anatomical structure of the brain, at least at the current spatial resolution.

The quantitative summary plots (Fig 2C) provide interesting insights. Anisotropic smoothing reduces variance but does not incur appreciable bias (arrow 1). In contrast, other methods such as Gaussian smoothing reduce variance but incur substantial bias (arrow 2). Thus, a bias-variance tradeoff does not necessarily occur in all situations. We also see that error is not a perfect metric to discriminate amongst methods, as both anisotropic smoothing (location 3) and Gaussian smoothing (location 4) yield comparable levels of correlation between analysis results and ground truth. Finally, there is a general relationship between reducing variance and increasing similarity to ground truth (arrow 5). This makes sense since denoising methods should, in theory, reduce unwanted measurement noise and generally push results towards the ground truth.

In the second simulation, we use as ground truth a synthetic hemodynamic response function (Fig 3A, top) and simulate noisy measurements of this ground truth by adding temporally correlated Gaussian noise. As expected, the raw data ('No denoising') follow the ground truth, in the sense of lacking bias, but suffer from high variance (Fig 3B, first column). The method of reconstructing the measurements using a small set of basis functions ('Basis restriction') greatly reduces variance but incurs major deviations from the ground truth (Fig 3B, second column). The discrepancy can be traced to the fact that the basis functions do not have much dynamics around the time of the undershoot (see blue arrow). The method of fitting a parametric function to the data ('Parametric fit') provides variance reduction and, appealingly, introduces very little bias, if any (Fig 3C, third column). This makes sense, since the parametric function used to fit the data is the same function that was used to generate the ground truth. If a different parametric function were used, these results of course may no longer hold.

The quantitative summary plots (Fig 3C) bear out the above observations. Basis restriction is very effective at reducing variance but is highly biased (location 1). Nonetheless, on balance, the bias-variance tradeoff is such that error is reduced compared to no denoising (location 2). However, there is even a better method: parametric fitting is essentially unbiased (location 3)

and performs the best at achieving results that are similar to the ground truth (location 4). Interestingly, even though parametric fitting has *more* variance across analysis results than basis restriction, parametric fitting yields results that better match ground truth (arrow 5). This can be understood as the consequence of the undesirable bias that is induced by basis restriction.

In the third simulation, we use as ground truth a synthetic set of tuning curves (10 units, 50 conditions) whose dimensionality is fixed to 4 (Fig 4A) and simulate noisy measurements of this ground truth by adding Gaussian noise. As expected, the raw data ('No denoising') follow the ground truth, in the sense of lacking bias, but suffer from high variance (Fig 4B, first column). The method of boxcar smoothing substantially reduces variance and, appealingly, does not incur any appreciable bias (Fig 4B, second column). This makes sense given that the width of the boxcar used is 3, which is relatively small compared to the intrinsic smoothness of the ground-truth tuning curves. The method of dimensionality reduction using principal components analysis (PCA) yields variance reduction at the expense of bias, with the specific bias-variance tradeoff controlled by the number of dimensions. Specifically, if dimensionality is aggressively reduced, more variance reduction is achieved but more bias is introduced (e.g., Fig 4B, sixth column). If dimensionality is reduced less aggressively, less variance reduction is achieved but less bias is introduced (e.g., Fig 4B, third column).

The quantitative summary plots (Fig 4C) provide additional insight. The bias-variance tradeoff in PCA is clearly visible: there is a continuous progression from PCA6 to PCA2 in terms of increasing amounts of bias and decreasing amounts of variance (arrow 1). Compared to PCA8, PCA6 does not incur appreciable bias; this suggests that preserving six dimensions is sufficient to retain all (or nearly all) of the underlying signal in the noisy measurements. The method of boxcar smoothing clearly outperforms PCA, as it greatly reduces variance and does not induce bias (location 2), and, moreover, achieves the best match to ground truth (location 3). Interestingly, the number of dimensions in PCA that maximizes similarity to ground truth is 3 (location 4), which is not the same as the true dimensionality of the underlying representation. This may seem counterintuitive at first, but can be understood as the simple consequence of the mixing of bias and variance when quantifying similarity to ground truth. That is, even though retaining only 3 dimensions is guaranteed to discard some of the true signal and incur bias (since the ground-truth dimensionality is 4), the reduction of variance afforded by retaining only 3 dimensions apparently improves the overall similarity to ground truth. Perhaps the most important insight is that reducing dimensionality to 4 already starts to introduce noticeable levels of bias (location 5). This is due to the fact that in the presence of measurement noise, the dimensions identified by PCA will start to deviate from the true dimensions that underlie the ground-truth representation. In other words, noise inevitably corrupts all of the PCA-identified dimensions, not just the ones that are discarded [6]. Hence, there is no guarantee that using 4 dimensions will retain all of the relevant signal contained in a given measurement.

## Downstream impact of denoising

The examples provided above demonstrate how bias can be formally studied in practical situations. However, users of denoising methods are probably not fundamentally interested in low-level data characteristics such as the amount of bias on individual data points, but are probably more interested in the impact that bias might have on downstream analysis results. To provide insight into this matter, we conduct an example downstream analysis in which we take simulations of noisy anatomical MRI data (as in Fig 2), and assess the quality of tissue segmentations obtained after applying different denoising methods (Fig 5).

As expected, segmentation results based on the raw data are poor, with numerous speckles and inaccurate labels of gray and white matter (Fig 5, second column). The atlas-based method improves the robustness of the segmentation, reducing speckles, but produces fairly inaccurate segmentation topology (Fig 5, fourth column). Simple Gaussian smoothing yields very robust results (Fig 5, third column), and in fact, the overall topology of the segmentation appears reasonably matched to the segmentation based on the ground truth (Fig 5, first column). Finally, we see that anisotropic smoothing produces excellent results. An important insight from these results is that the quality of downstream analysis results for different denoising methods may not necessarily mirror the performance of these methods on low-level data metrics. For example, from the point of view of bias, Gaussian smoothing seems quite undesirable (see Fig 2C), but from the point of view of tissue segmentation, the results based on Gaussian smoothing are actually quite respectable. Thus, the decisions that one makes regarding denoising methods should take into account not only the potential impact on low-level data metrics like bias, but also the larger goals that one has for a set of data.

## Discussion

In this paper, we have described a simple framework for evaluating denoising methods, and we have provided examples that highlight important (and possibly surprising) observations about denoising. These examples were not intended to benchmark the performance of state-of-the-art methods, but rather to demonstrate insights into the nature of denoising. The main issues that we focus on—bias and variance—are well-understood in statistics. We believe these issues need increased attention in experimental fields, especially in light of the increasing complexity of datasets and analysis pipelines. While developing denoising techniques to improve data quality is a worthwhile endeavor, we should approach such techniques with caution and strive to avoid introducing systematic bias to our measurements (see also the perspective by [29]). To summarize our viewpoint, we propose the following three action items.

### We should acknowledge bias

As a first action item, we should acknowledge bias as a major potential concern when applying denoising methods. When making measurements, a presumption is that repeated measurements will help the researcher narrow the range of plausible values for the parameter of interest. In this context, systematic bias should be alarming. Some denoising methods might not introduce bias, and it might be possible to see that this is the case from a theoretical perspective. However, in general, denoising methods are likely bound to the bias-variance tradeoff: there is likely going to be a tradeoff between reduction of variance and introduction of bias. Even if one does not yet know exactly what the bias is for a given method, it is worthwhile to acknowledge and discuss what this potential bias might be. In a sense, it should not be surprising that bias should be a potential concern with denoising methods. Indeed, when presented with a denoising method, it is common to hear the reaction "How do you know you aren't removing signal?", which can be viewed as an informal expression of the issue of bias.

An intuitive way to think about bias is through the concept of a prior. Denoising methods can be viewed as bringing priors to a set of data [12]. On the one hand, if we do not incorporate any priors, the data in their raw form are noisy but safe: they can be expected to provide the right answer on average (assuming that the noise is zero-mean). On the other hand, if we apply a denoising method, we are bringing in priors, or implicit assumptions, regarding the nature of the underlying system. The key question is whether the priors embodied by the denoising method are a good match to the system. If the priors are very well matched (e.g. Fig 2B, fourth column), little or no bias is introduced, and we can enjoy the reduced variance. If

the priors are not well matched (e.g. Fig 3B, second column), bias is introduced, and the reduced variance may not be worth it. Whether a given denoising method is well matched to a system may vary across situations. For example, anisotropic smoothing (e.g. Fig 2B, fourth column) is likely inappropriate for structures consisting of point-like features; Gaussian smoothing (e.g. Fig 2B, second column) is actually a good approach in situations where the measurement resolution is high compared to the scale of the underlying signal of interest. Analogous to the "No Free Lunch Theorem" in optimization [30], we should recognize that any given denoising method is not guaranteed to perform well in all situations. Accordingly, our goal should not only be to demonstrate that a given denoising method performs well in certain situations, but should also be to identify the range of situations within which the method performs well and the range within which the method fails.

Illustrative examples of priors come in cases where there are literally no data. These cases conveniently expose the full nature of the prior embodied by a technique. For instance, suppose we delete a small region of a photograph and use an image inpainting technique to fill in the region. While we are likely to obtain a reasonable-looking image that generally conforms to natural image statistics, it is obvious that this is no substitute for actual measurement. Had there been a specific object in the deleted region, it is likely that the inpainting technique would miss this completely and instead fill the region with general texture priors [31]. In other words, the technique would likely incur massive bias. Or, as a different example, suppose we train a model to predict high-resolution details that typically accompany low-resolution measurements. This model might be quite effective within a certain data regime at predicting high-resolution details when only low-resolution data are available, but might make non-sensical predictions when exposed to novel data regimes that differ substantially from the training dataset [29].

## We should study and quantify bias

As a second action item, we should study the bias that may be present in a denoising technique, and quantify its magnitude in real-world situations. Carefully characterizing the bias of a method is useful for providing full transparency and enabling accurate risk assessment. Bias can be studied using different approaches. It might be possible to make a theoretical assessment as to whether a denoising technique is likely to incur bias and what this bias might be like. This is feasible for denoising techniques that are based on simple, clear principles. For example, although simple smoothing is a naive approach, one appealing feature is that we fully understand the risk of bias that it entails. In contrast, denoising techniques that derive their power from large amounts of training data (e.g. deep neural networks) or techniques that derive noise estimates from the data themselves are more difficult to assess from an *a priori* perspective. Alternatively, we can use empirical analyses to evaluate the bias in denoising techniques (like the examples shown in Figs 2–4).

One of the take-home points of this paper is that different denoising metrics provide fundamentally different information:

- *One metric of denoising performance is variance.* Examining variability of results across repeated measurements or independent splits of a dataset provides useful information. All else equal, we want less variance.

- *A second metric of denoising performance is error.* Assessing error is a widely used approach in image processing [32, 33] where one seeks to minimize the error between a denoised output and a reference ground-truth image. Error can be quantified in various ways, such as mean squared error, peak signal-to-noise ratio, or structural similarity index [34, 35].

Alternatively, error can be assessed through the use of cross-validation to assess generalization to unseen samples, which serve as an implicit ground truth. All else equal, we want less error.

- *A third metric of denoising performance is bias.* Bias can be quantified by applying a denoising method to multiple independent measurements and carefully comparing the mean of the results to a ground-truth measure (as shown in Figs 2–4). All else equal, we want less bias.

It is clear that variance, in and of itself, is an inadequate denoising metric since it is unaffected by (and therefore does not assess) bias. However, since error reflects the combined influence of bias and variance, could it be a good policy to use error as a denoising metric? Indeed, some perspectives imply that error is the ultimate criterion and anything that reduces error is desirable [26, 36]. While we acknowledge that error is an extremely useful metric, we believe that it is valuable to isolate and quantify bias *in addition to* error. It is only by isolating bias that we can understand its prevalence and what downstream impact it may have on inferences made from a set of data. We make this suggestion under full acknowledgment that we ourselves have not fully implemented these ideas in the past. For example, we used cross-validated error to evaluate denoising performance in this study [37], but it would have been even more informative had we specifically assessed bias.

Although we demonstrate ground-truth simulations in this paper, studying bias is not limited to such situations. On the one hand, ground-truth simulations can deliver many valuable insights [17, 18]. However, ground-truth simulations are susceptible to the criticism that they may not capture the full complexity of real empirical data. Fortunately, it is possible to study bias in real data if one has access to a dataset in which many repeated measurements are available. One approach is to average across these measurements, treat the result as ground truth, and evaluate how well denoising methods can use single (or a few) measurements to recover the ground truth. Note that perfect recovery is not necessarily desired in this scenario since the ground-truth measure is still subject to some amount of measurement error.

Denoising efforts, especially in the field of machine learning, often place great emphasis on improving predictive performance, in the sense of generating results that better approximate a target ground-truth measure. While this engineering mindset has obvious practical and commercial value [38] and can be quite effective in driving competition and therefore progress [39], it falls short as a means for assessing measurement accuracy. Specifically, *predictive performance reflects a combination of bias and variance and therefore is insufficient in and of itself for studying bias*. Unless predictive performance is perfect, there is a potential that bias exists for a given denoising technique. Emphasis on prediction can be viewed in terms of the divide between what has been termed 'predictive modeling' and 'explanatory modeling': "predictive modeling seeks to minimize the combination of bias and estimation variance, occasionally sacrificing theoretical accuracy [i.e. correct identification of properties of the underlying system] for improved empirical precision" [40].

## We should consider the risk of bias to one's goals

As a third action item, we should consider the risk of bias in the context of the broader goals of a given endeavor. All else being equal, we would argue that for everyday scientific measurements, we cannot risk using denoising methods that introduce bias, as this may lead to incorrect inferences from the data. However, adopting a more realistic perspective, we recognize that the bias that might be present in a given denoised dataset could be a relatively minor aspect of the data. Even if we know with certainty that a denoising method introduces bias, we

might reasonably ask: how strongly does the bias affect the main issues at stake? We think that the best strategy is to consider each situation on a case-by-case basis and make a deliberate decision regarding the risk of bias.

*In some situations, bias might be acceptable.* For instance, if the goal is to clean an audio or video signal for aesthetic purposes or for basic perceptual interpretation [34], then bias would seem to cause little harm and a reasonable stance is to simply resign and accept bias [32]. An example of this is a clinician who is visually inspecting an image. If a denoising method incurs a little bit of blurriness, this does not seem to pose a major problem (assuming the clinician is aware of the blurriness). Or, if a denoising method affects an aspect of a given dataset that does not have substantial impact on the main findings from the data (e.g., small changes in region identification might be unlikely to change the overall measured activity from a brain region), then bias would not seem to be a problem.

*In other situations, bias might be unappealing but must be accepted out of necessity.* For example, if a dataset is too noisy to make inferences and additional measurements are not possible (e.g., due to the rarity of the data), it may be necessary to apply a denoising method in order to salvage the data and make some inferences, even if imperfect. Alternatively, it might be the case that implementing an unbiased analysis method might require an inordinate amount of time (either human time or computational time). In such cases, the user might need to use biased analysis methods out of practical necessity.

*But in many situations, bias may be unacceptable.* For example, if a set of noisy data is being used to make a clinical diagnosis, it might be better to leave the data untouched and acknowledge that the data are inconclusive than to risk introducing an artifact or removing a true signal. Or, as another example, if a set of data is intended to critically test hypotheses about temporal characteristics of a system, one might avoid applying a denoising method that has access to multiple temporal measurements, as the method might potentially introduce biases in the temporal domain, and instead restrict the method to single measurements at a time.

## Concluding remarks

In this paper, we have emphasized bias as an important property that should be considered when evaluating denoising methods. In practice, how might one select which of several denoising methods to use? One approach is to establish a data regime for which one would like a method to perform well, determine which methods are unbiased (or nearly unbiased) in this regime, and then select from these methods the one that has the least variance. However, bias is just one of several factors that influence the larger goals for a set of data. As discussed above, bias might not have a major impact on a user's end goals and the inferences that they wish to make. Additionally, there are other important considerations to take into account when considering an analysis method. These include the availability of a working implementation, the time required to incorporate a method into an existing pipeline, execution time, robustness across diverse types of data, and the clarity and interpretability of the procedures that underlie the method. Thus, our broader position is that the possibility of bias should be one of many factors that enters a user's informed decision regarding the specific methods to apply to a set of research data.

## Acknowledgments

We thank M. Akcakaya, T. Coalson, M. Glasser, O. Gulban, T. Knapen, E. Merriam, A. Rokem, and J. Winawer for helpful discussions and comments on the manuscript.

## Author Contributions

**Conceptualization:** Kendrick Kay.

**Formal analysis:** Kendrick Kay.

**Investigation:** Kendrick Kay.

**Methodology:** Kendrick Kay.

**Software:** Kendrick Kay.

**Validation:** Kendrick Kay.

**Visualization:** Kendrick Kay.

**Writing – original draft:** Kendrick Kay.

**Writing – review & editing:** Kendrick Kay.

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
