## [Decision Letter · Decision Letter 0]

5 May 2022

PONE-D-22-05034The risk of bias in denoising methodsPLOS ONE

Dear Dr. Kay,

Thank you for submitting your manuscript to PLOS ONE. After careful consideration, we feel that it has merit but does not fully meet PLOS ONE’s publication criteria as it currently stands. Therefore, we invite you to submit a revised version of the manuscript that addresses the points raised during the review process.

We look forward to receiving your revised manuscript.

Kind regards,

Pew-Thian Yap

Academic Editor

PLOS ONE

Journal Requirements:

2. Please note that PLOS ONE requires that experiments, statistics, and other analyses must be performed to a high technical standard and described in sufficient detail to allow for reproducibility of the study (http://journals.plos.org/plosone/s/criteria-for-publication#loc-3). To demonstrate the performance of the tool, we would expect comparisons to be drawn between existing state-of-the-art methods. Please clearly report at the beginning of your methods or results section which the key performance measures were to establish validity and utility of your method. Please also report clearly which statistical analysis was used to establish robustness of performance measures.

Reviewers' comments:

Reviewer's Responses to Questions

**Comments to the Author**

1. Is the manuscript technically sound, and do the data support the conclusions?

Reviewer #1: Yes

Reviewer #2: Yes

2. Has the statistical analysis been performed appropriately and rigorously? 

Reviewer #1: Yes

Reviewer #2: Yes

3. Have the authors made all data underlying the findings in their manuscript fully available?

Reviewer #1: Yes

Reviewer #2: Yes

4. Is the manuscript presented in an intelligible fashion and written in standard English?

Reviewer #1: Yes

Reviewer #2: Yes

5. Review Comments to the Author

Reviewer #1: The author highlights the bias introduced by denoising and that lower MSE with respect to the ground truth does not always mean lower bias. Overall, the manuscript was well written with comprehensive demonstrations of the bias after denoising with different methods.

My main concerns are

1. Is bias bad? Unbiased estimator does not always exist and sometimes biased estimator can lower MSE [1,2,3] (similar to Fig. 1, top right). While Fig.3 clearly indicates bias changes the shapes of the estimated HRF, Fig. 2 is not as clear. Gaussian smoothing introduced larger bias compared to anisotropic smoothing but is it bad and how it would affect, for example, clinical diagnosis? Except from the fact the Gaussian smoothing images are blurry (which is uncorrelated to bias), it is still unclear how the bias is dangerous in this case. Perhaps showing an example of bias can obscure a structure of interest (tumor) would highlight the point.

2. How bias affect downstream analyses? Continuing point 1 above, some subsequent analyses should be done to highlight how bias would affect the results and how severe it could be. In the context of MRI in this paper, it could be segmentation from sMRI, correlation between stimuli and signal from fMRI, or tensor fitting and tractography from dMRI. Without subsequent analyses, although we acknowledge there is a bias, it is still unclear whether we should spend effort removing it with the cost of increasing variance (e.g., using parametric fit instead of basis restriction in Fig.3).

3. Is MSE still a good performance measurement? It is obvious that smaller bias and variance are favored. But in case of a trade-off between variance and bias like many methods shown in this work, is measuring bias alone a good practice? MSE accounts for both bias and variance and is a good measure to balance out the two terms, especially when to determine in a decrease in bias/variance worth it (given an increase in the other one). Again, this highlights the importance of demonstrating how bias would affect the images visually and subsequent analyses. If it does not affect those significantly, it might be sufficient to quantify MSE alone.

Minor comment:

While the title is "The risk of bias in denoising methods", 2 out of 3 experiments are MRI-related. The title and the abstract should make clear that the manuscript experiments are limited to MRI data.

Although experiments are meant to demonstrate how bias would be introduced, the paper is undermined that little to no state-of-the-art denoising method was used (while they are mentioned and discussed many times throughout the manuscript).

[1] Eldar, Yonina C. Rethinking biased estimation: Improving maximum likelihood and the Cramér-Rao bound. Now Publishers Inc, 2008.

[2] Chatterjee, Priyam, and Peyman Milanfar. "Is denoising dead?." IEEE Transactions on Image Processing 19.4 (2009): 895-911.

[3] Kay, Steven, and Yonina C. Eldar. "Rethinking biased estimation [lecture notes]." IEEE Signal Processing Magazine 25.3 (2008): 133-136.

Reviewer #2: This work studies the risk of bias in denoising methods. More specially, denoising methods may introduce bias, which has risks leading to incorrect scientific inferences. To investigate the denoising-induced bias, three simulation experiments are performed, which draw three key conclusions and the importance of quantifying the possibly denoising-induced bias. The paper is well written and easy to follow. The conclusions from this work can benefit a large number of denoising works and provide valuable guidance for the use of denoising methods. Regarding this work, I only have several minor concerns.

1. The first simulation is based on Gaussian noise corrupted MR images. However, depending on the k-space data acquisition schemes and reconstruction approaches, the noise in MR images follows a stationary/non-stationary Rician/non-central Chi noise [1,2]. I am wondering whether the Gaussian noise assumption has some influences on the experimental results.

2. It is also helpful to provide some insightful discussions on how to define a good denoising method, which can benefit the medical image computing community.

[1] Statistical Analysis of Noise in MRI.

[2] Denoising of Diffusion MRI Data via Graph Denoising of Diffusion MRI Data via Graph Framelet Matching in x-q Space

6. PLOS authors have the option to publish the peer review history of their article (what does this mean?). If published, this will include your full peer review and any attached files.

Reviewer #1: No

Reviewer #2: No

---

## [Author Response · Author response to Decision Letter 0]

10 May 2022

The full response-to-reviewers letter is attached to this submission.

---

## [Decision Letter · Decision Letter 1]

20 Jun 2022

The risk of bias in denoising methods: examples from neuroimaging

PONE-D-22-05034R1

Dear Dr. Kay,

We’re pleased to inform you that your manuscript has been judged scientifically suitable for publication and will be formally accepted for publication once it meets all outstanding technical requirements.

Kind regards,

Pew-Thian Yap

Academic Editor

PLOS ONE

Additional Editor Comments (optional):

Reviewers' comments:

Reviewer's Responses to Questions

**Comments to the Author**

1. If the authors have adequately addressed your comments raised in a previous round of review and you feel that this manuscript is now acceptable for publication, you may indicate that here to bypass the “Comments to the Author” section, enter your conflict of interest statement in the “Confidential to Editor” section, and submit your "Accept" recommendation.

Reviewer #1: All comments have been addressed

Reviewer #2: All comments have been addressed

2. Is the manuscript technically sound, and do the data support the conclusions?

Reviewer #1: Yes

Reviewer #2: Yes

3. Has the statistical analysis been performed appropriately and rigorously? 

Reviewer #1: Yes

Reviewer #2: Yes

4. Have the authors made all data underlying the findings in their manuscript fully available?

Reviewer #1: Yes

Reviewer #2: Yes

5. Is the manuscript presented in an intelligible fashion and written in standard English?

Reviewer #1: Yes

Reviewer #2: Yes

6. Review Comments to the Author

Reviewer #1: (No Response)

Reviewer #2: (No Response)

7. PLOS authors have the option to publish the peer review history of their article (what does this mean?). If published, this will include your full peer review and any attached files.

Reviewer #1: No

Reviewer #2: No

---

## [Editor Report · Acceptance letter]

23 Jun 2022

PONE-D-22-05034R1 

The risk of bias in denoising methods:
examples from neuroimaging 

Dear Dr. Kay:

I'm pleased to inform you that your manuscript has been deemed suitable for publication in PLOS ONE. Congratulations! Your manuscript is now with our production department. 

Kind regards, 

on behalf of

Dr. Pew-Thian Yap 

Academic Editor

PLOS ONE